# Library of Apollo: A Virtual Library Experience in your Web Browser

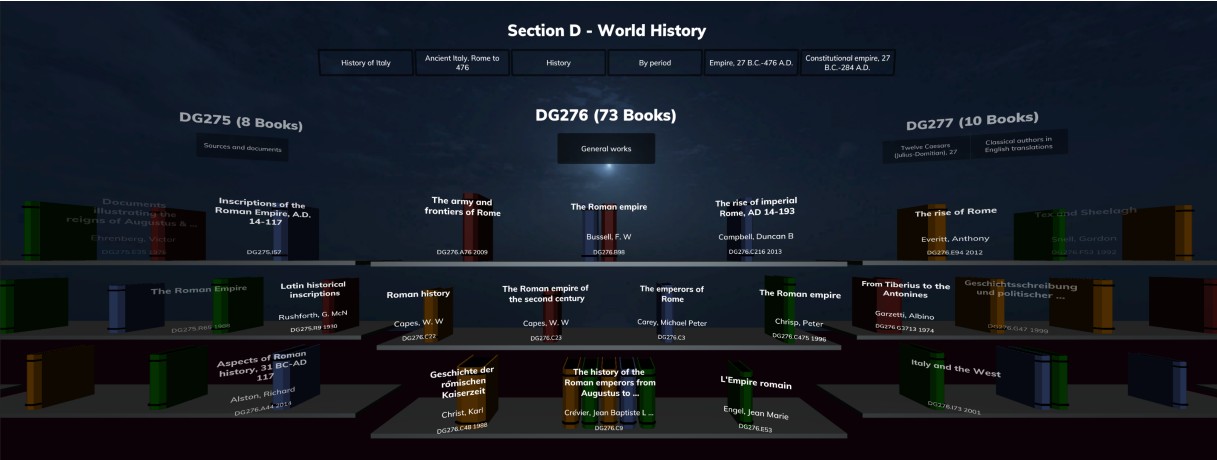

Figure 1: A wide-angle shot of the shelves. The user can scroll the shelves left and right, look around using their mouse and click & engage with the books, signage and panels.

## ABSTRACT

Research libraries that house a large number of books organize, classify and lay out their collections by using increasingly rigorous classification systems. These classification systems make relations between books physically explicit within the dense library shelves, and allow for fast, easy and high quality serendipitous book discovery. We suggest that online libraries can offer the same browsing experience and serendipitous discovery that physical libraries do. To explore this, we introduce the Virtual Library, a virtual-reality experience which aims to bring together the connectedness and navigability of digital collections and the familiar browsing experience of physical libraries. Library of Apollo is an infinitely scrollable array of shelves that has 9.4 million books distributed over 200,000 hierarchical Library of Congress Classification categories, the most common library classification system in the U.S. The users can jump between books and categories by search, clicking on the subject tags on books, and by navigating the classification tree by simple collapse and expand options. An online deployment of our system, together with user surveys and collected session data showed that our users showed a strong preference for our system for book discovery with 41/45 of users saying they are positively inclined to recommend it to a bookish friend.

**Index Terms:** Human-centered computing——
—

## 1 INTRODUCTION

Book exploration is a fundamental part of every reader's life. Libraries often play a unique role in this exploratory process. Large collections are classified, organized and laid out through the use of continuously-updated, rigorous and systematic classification systems, such as the Library of Congress Classification (LCC) system in the U.S. Under the LCC system, books are uniquely classified into a tree of classifications, and are additionally tagged with one or more standardized subject headings.

These categorization systems make relationships between books physically explicit, putting each book in the context of every other book within the same and nearby shelves that have similar books, adjacent in subjects, authorship, time and geography [1,2]. This allows for natural, fast and high quality serendipitous book discoveries.

Online library interfaces are generally search result based and this takes away from the valuable serendipitous discoveries that are often made in physical libraries through browsing [3–5]. In this paper, we introduce a Virtual Library implementation, Library of Apollo, which aims to bring together the connectedness and navigability of digital collections and the familiar browsing experience of physical libraries. Library of Apollo is an infinitely scrollable array of bookshelves that has 9.4 million books distributed over 200,000 hierarchical Library of Congress Classification (LCC) categories. The users can jump between books and categories by targeted search, clicking on the subject tags on books, and by navigating the classification tree by simple collapse and expand options.

The main contributions of this paper are two-fold. Firstly, we contribute the design and implementation of the Virtual Library system, which replicates a physical library's collection outline and categorization features while adapting it to a web browser setting that is controlled via mouse and keyboard. We discuss the improvements we made to increase the internal cross-connectivity, ease-of-use and navigability, along with the technical details necessary to serve a large and organized collection of books virtually. Second, we deployed this system publicly on the Internet at loapollo.com, soliciting survey feedback from visitors. Our analysis of the survey data and server logs reveal changing behaviours around physical libraries and bookstores due to the ongoing COVID-19 pandemic, and their preferences for using the Virtual Library system.

## 2 RELATED WORK

### 2.1 Library of Congress Classification System

There are different classification systems used for different purposes in different parts of the world, and as we have focused on large collections as those hosted by research and academic libraries, we

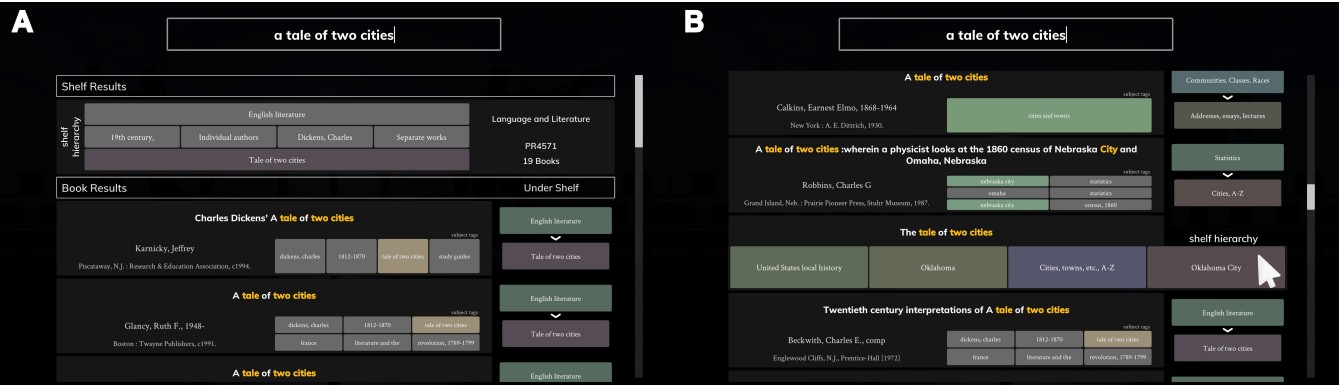

Figure 2: Search results for 'a tale of two cities', showing both shelf and book results. (A) Shelf results list LCC classes that are search-hits, while book results are hits for titles, authors and subject headings of books. The terms that led to the search-hits are highlighted. (B) Hovering over the 'Under Shelf' portion of book results expands the shelf hierarchy.

decided to use the subject-based, hierarchical Library of Congress Classification system (LCC) [1], currently one of the most widely used library classification systems in the world [6].

LCC divides all knowledge into twenty-one basic classes, each identified by a single letter. These are then further divided into more specific subclasses, identified by two or three letter combinations (class N, Art, has subclasses NA, Architecture; NB, Sculpture etc.). Each subclass includes a loosely hierarchical arrangement of the topics pertinent to the subclass, going from the general to the more specific, often also specifying form, place and time [1,6]. LCC grows with human knowledge and is updated regularly by the Library of Congress [6].

## 2.2 Book Browsing and Serendipity in Libraries

Information seeking in physical libraries takes two forms: search of known items and browsing. Patrons may begin with a search of a known item, but through undirected browsing of nearby items, often discover new and unknown books serendipitously [7]. Even though patrons could conceivably find interesting items in completely disarranged stacks, libraries aim to ease this fundamental browsing process by cataloguing, classifying and shelving books according to some classification system [2], putting like near like, deemed so through their topics, subtopics, authorship, form, place and time.

Library shelves are meticulously organized to encourage serendipity [7, 8] and call numbers themselves are markers that indicate the semantic content of shelved items [9]. There is some quantified evidence about the neighbour effects created by Dewey Decimal and Library of Congress Classification systems: a nearby book being loaned increases the probability of a subsequent loan by at least 9-fold [10], indicating that browsing is happening regularly and effectively within these classification systems. There is also strong spoken and anecdotal preference for physical library shelf browsing [3, 11] and users bemoan the lack of opportunity to do so online [3–5].

More recently, McKay et. al. detailed the actions taken during library browsing by users, like reading signs, scanning, the number of books, shelves, bays touched and examined etc., indicating an idiosyncratic browsing behaviour by each user as well as a high success rate, with 87% of users leaving their browsing session with one or more books [12]. These results suggest a unique and creative character to browsing within the general scope of information retrieval.

## 2.3 Book Browsing and Serendipity in the Digital Age

Even though libraries have been amongst the earliest adopters of computers, digitization and online access [13] and there has been a

clearly expressed sentiment favoring browsing [3–5, 11], there has been little commensurate effort to carry the physical shelf browsing experience to online libraries. Online library interfaces are almost universally search-result based, and we know the vast majority of users only see up to ten search results [14], and only interact with one or two in any depth for further information [15]. These small numbers are hardly conducive to browsing and serendipitous discovery. The recommender systems used by online vendors are generally good to discover popular books, popular tastes and genres while they often fail to provide novel ranges of selections necessary for browsing and serendipity [16].

There is some novel recent work that has been developed to facilitate serendipitous browsing in libraries [17, 18]. The Bohemian Bookshelf [17] aimed to encourage serendipity by creating five interlinked visualizations over its collection, based on books' covers, sizes, keywords, authorships and timelines. This playful approach was deployed on a touch kiosk in a library and was received very positively, but the collection size used for implementation and test was limited to only 250 books, a number too small to be indicative of performance over larger collections where some of the employed visualizations could become too cluttered and complex to navigate.

The Blended Shelf [18] carried over the physical shelves into very large touch displays that offered 3D visualizations of a library collection of 2.1 m books, conforming to the standard classification used by that library. The 3D shelves were draggable by swipe, can be searched and reordered and the classifications can be navigated via a breadcrumb-cued input mechanism. However there are no deployments, tests or user studies regarding their implementation.

We have taken a reality-based presentation approach, creating a 3D online library of 9.4m books on infinitely scrollable shelves that conform to the LCC ordering, that is also navigable through search, cross-connected subjects, and a navigatable classification tree.

## 3 DESIGN AND IMPLEMENTATION

We have regarded the strongly expressed preference for physical shelf browsing [3–5, 11] together with the long cultivated art of library classification systems [1] as the guiding principles of our design process, and aimed to bring the physical library experience online while making improvements to enhance the connectedness and navigability of the served online collection.

### 3.1 Designing to Faithfully Recreate Physical Libraries

We had two conflicting design interests. First, we wanted to design and deploy a virtual library application at scale available to everyone on web browsers. Second, we also wanted this to be as close to a physical library experience as possible. Naïvely optimizing for the

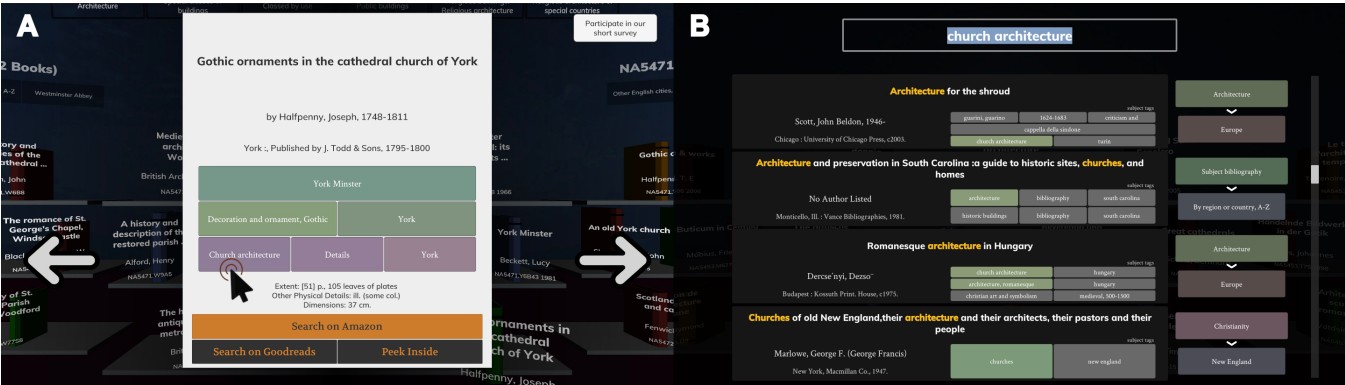

Figure 3: (A) Clicks on shelved books bring up synoptic book panels. The elements in the subject heading table are clickable. (B) The search-results after the "church architecture" subject heading was clicked. The hits on subject headings are highlighted with shades of green.

latter would have meant building an online replica of a huge library in 3D, that would then be navigated by mouse and keyboard through some combination of user motion and teleportation. However, that would have been an alien set of interactions to most people, be hard to navigate and possibly even clunky.

Instead we extracted the most crucial part of the physical book browsing experience from the physical library: the shelves. The user, essentially the camera, is placed in front of a set of floating shelves, and is allowed 20 degrees of freedom of camera motion left and right, and 10 degrees up and down, achievable with a mouse or a trackpad. A wide-angle view of the shelves is seen on Figure 1. The

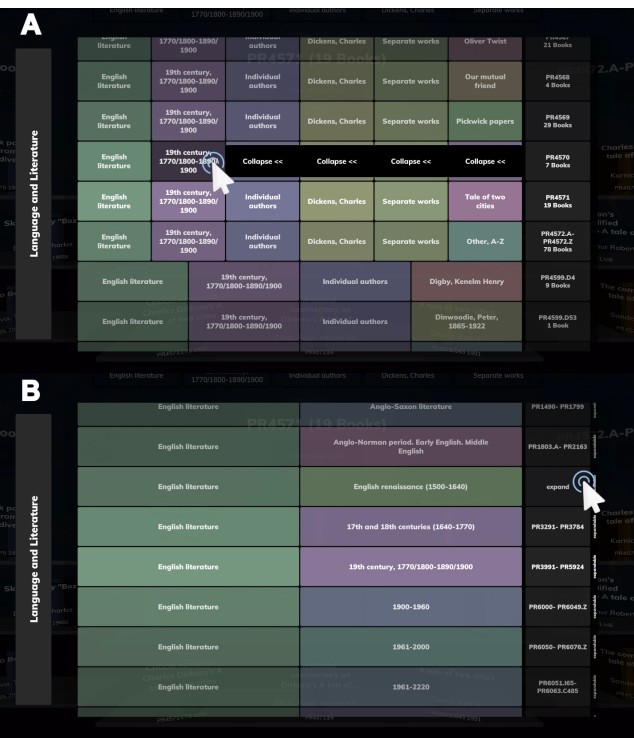

Figure 4: (A) The scroll-view of class hierarchies. Hierarchies can be collapsed by clicking any of the intermediate elements. (B) The collapsed hierarchies after the click at A. The hierarchies can be scrolled up and down, or expanded by clicking their tag panels if they contain child categories.

user can "scroll" the shelves left and right as slow or as fast as they want via the regular scroll gesture on their trackpads, scroll-wheels on mouses or arrow-keys on keyboards (Video Figure).

Books on the virtual shelves are arranged in order according to their LCC classifications, just as they would be in a real library. There are also panels showing LCC hierarchies above shelves indicating where the user is within the library. The panels in the center indicate the user's current location, while the left and right panels respectively indicate the previous and next classifications, and thus the content of the shelves that await the user in those directions. A click on the left or right panels scroll the shelves to bring those classes in front of the user. A click on the center panels opens the LCC classification hierarchy as seen on Figure 4. This list can be navigated by up and down scrolling as well as clicks to expand and collapse it. Clicking on the end portion of these hierarchies brings that class & its associated books in front of the user.

The user can start typing anytime they want to bring up the search bar, which can be used to search over titles, authors, subjects and LCC classifications. If any book or classification is selected through the search results, the user is transported to the shelf containing this selected book. The books are also sized according to their real dimensions, page numbers and volumes, as seen on Figure 1.

### 3.2 Building a Familiar and Expressive Search Bar

Single input search bars are ubiquitous and users are intimately familiar with them. We utilized the search bar as an entry point into our library by designing it as a catch-all input system that searches over book titles, authors, subject headings of books, and LCC categories. Our search provides a way for users to enter into the hierarchically organized stacks by helping them first with identifying a relevant pool of items they might be interested in.

Figure 2 shows shelf (class) and book search hits for the query 'A tale of two cities.' Shelf results show the LCC hierarchy from top to bottom, together with the class numbers and the number of books that are listed under that class. Book results list titles, authors and publishers, together with a table showing the subject headings listed for each book. To the right of each book result, you can also see under which shelf (class) that book can be found, hovering over that section expands the shelf hierarchy as can be seen on the right side of Figure 2. The 'under shelf' portions are always colored to clearly designate shelf hierarchies. The search results are not paginated, and can be "infinitely" scrolled downwards until the end of results.

### 3.3 Engineering a Smooth Browsing Experience

Clicking on a search result, both for books and shelves, zooms the user into a set of floating shelves (Figure 1). The searched-for book appears on the centre shelf and is briefly highlighted as an indicator

of its position. The books on shelves conform to LCC sorting and appear as they would on a regular library. One other key feature we have added is browsing over the LCC classification hierarchy as seen on Figure 4. This scroll-view of classification hierarchies is opened by clicking on the centered LCC panels. A listed hierarchy can be expanded or collapsed by clicking on different parts of the hierarchy. Clicking on any panel within a hierarchy collapses that hierarchy down to that panel, reducing the number of hierarchies in the scroll-view. Clicking on the name-tag panel of a hierarchy expands that hierarchy. Clicking on the last panel of a hierarchy transports the user to the beginning of that hierarchical class with the first book of that class highlighted on shelves.

Another addition to the browsing experience is through the improved connectivity that comes through search over subject headings of books. Book clicks bring up synoptic windows as seen on part 1 of Figure 3. The colored subject tables are filled with Library of Congress Subject Headings (LCSH) which we have populated for every book from a dataset maintained by domain experts at the Library of Congress [1]. These LCSH headings are clickable and trigger a search over the entire dataset's indexed LCSH fields. Part 2 of Figure 3 shows search results for an LCSH field search; notice that these books can come from entirely different shelves and classes, often very distant from each other, thus providing another way to jump between classes and books. This allows users to transcend the distance imposed by LCC in physical libraries through a subject-based search method, which allows for dynamic links between books. We believe this feature would go a long way to satisfy the expressed desire by users to occasionally see distant books [5].

### 3.4 Providing Access to Digitized and Physical Books

Another killer feature of real world libraries is in-depth browsing of individual books. A library-goer can just pick up any book and read until their curiosity is sated. In order to provide a similar experience, when a user clicks on a shelved book, we provide a single page overview. Part 1 of Figure 3 shows an example of this page which displays the book's title, authorship, publishing house, colored subject headings, physical information regarding extent and dimensions, and links to lookup the same book on Amazon and Goodreads, as well as a Peek Inside button that is enabled when there is a free & digitized version available on OpenLibrary (which houses over 3 million books, or around 30% of the total collection). A single click on this button opens a new tab with an online reader showing the contents of the book, as seen on Figure 5, providing an in-depth reading experience. The Amazon and Goodreads buttons provide easy access to purchase and social reading options, respectively.

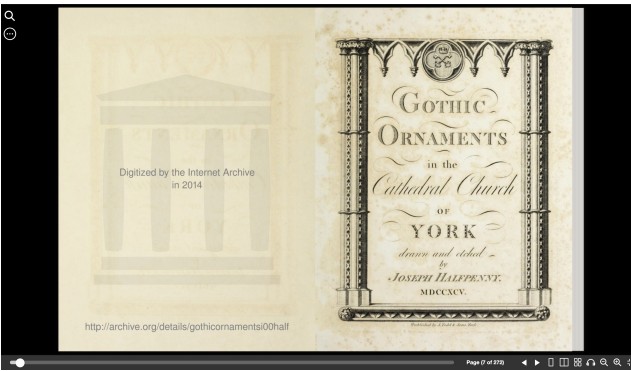

Figure 5: The "Peek Inside" button on book panels opens an e-reader that uses OpenLibrary's digitized book archive.

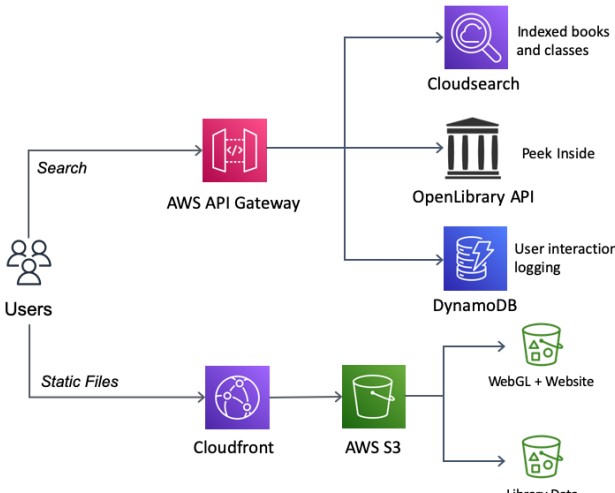

Figure 6: The cloud architecture of Library of Apollo.

## 4 ONLINE DEPLOYMENT AND SURVEY RESULTS

### 4.1 The Front-end and the Cloud Architecture

We have developed the front-end of our application using Unity WebGL. The compiled WebAssembly, Javascript and HTML files are hosted on the AWS S3 service through the CDN service of AWS, Cloudfront. The books are sorted according to LCC sorting scheme [1, 6] and clustered into JSON files that contain 500 books each, gzipped and stored in S3. These data files are fetched when the user is browsing the shelves, and pre-fetched during scrolling when the user is near the end of a cluster.

The same data that is stored in S3 is indexed on AWS Cloudsearch to power the search functionalities of our app. All search requests are performed through the REST API we have developed on AWS API Gateway. Search results are populated through the data returned from Cloudsearch. For analytics, user click-data is also recorded on DynamoDB through the same gateway. The search requests to OpenLibrary's digitized books dataset are made in a similar fashion. Our architecture is summarized on Figure 6.

### 4.2 The Deployed Dataset

Courtesy of the Library of Congress [19], through their MARC Open-Access program, we were able to put together 9.4m books, complete with their LCSH and LCC data, distributed over 200k LCC classes. We pre-processed this data to serve our specific needs, and stored and indexed them for use within our virtual library application as described above. Despite the size of the dataset, the website is very smooth to use and inexpensive to operate - monthly operational costs are less than $40 USD.

### 4.3 Deployment and Recruitment

We deployed our library at a publicly accessible address, loapollo.com, and seeded links to the project via Reddit and word-of-mouth. Over a span of two weeks, over two hundred unique users visited the site. Each user to the library was assigned a randomly-generated persistent user ID, to track repeated session visits, as well as a session ID for any given visit. The Library front-end automatically logged click interactions with the system in a DynamoDB server for later introspection.

Of the 224 unique visitors, 21 (9.4%) visited more than once, with one user visiting the site a total of five times over the course of two weeks. Users interacted for an average of 162 seconds, and

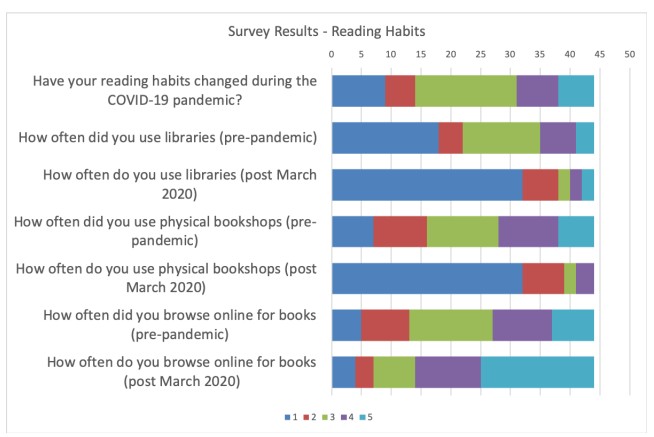

Figure 7: The change in reading habits due to COVID-19 pandemic.

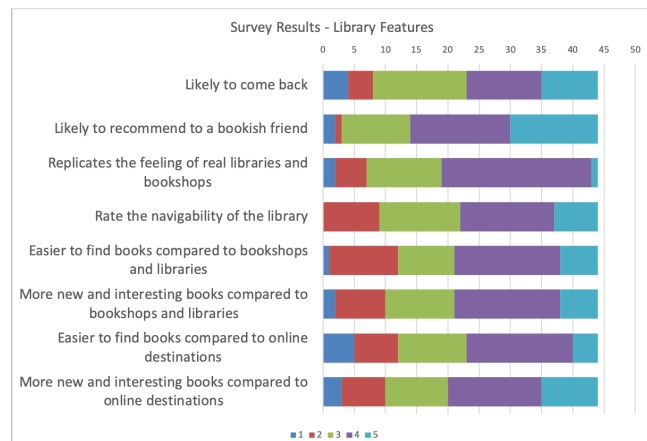

Figure 8: The user perception of the library's navigability, ease of use, browsing quality and overall quality.

produced an average of 8 click interactions; notably, however, both distributions are particularly long-tailed. While some users only visited for very brief moments (often just a single search, followed presumably by some scrolling), 12 users used it for over ten minutes each, and likewise 20 users generated over 20 click events each.

### 4.4 Survey

When users clicked on any book, a survey link was displayed in the upper-right corner, ensuring that users interacted with the system to a minimal extent prior to taking the survey. Users who accessed the link were invited to fill out a short survey with two major parts: a first section which asked about their existing (pre- and post-pandemic) book browsing habits, and another section with which asked about their experiences with the library. Finally, users could provide open-ended feedback about their experience with the Library. The full survey design can be found in the Supplemental Materials.

We received a total of 46 survey responses. Two responses were discarded: one filled out '1's for every single question, and another was a clear duplicate submission. Participants reported reading an average of 13.4 books per year (SD 15.2; participants reported anywhere from 1 book per year to 80 books per year).

In the first part of the survey (Figure 7), the participants did not report significantly changing their reading habits (mean 2.9), but evidently had a significant shift in book-browsing habits due to the pandemic. Library usage decreased significantly (pre-pandemic mean 2.36, post-pandemic mean 1.54), as did bookshop usage (pre-pandemic mean 2.98, post-pandemic mean 1.45), whereas online browsing significantly increased (pre-pandemic mean 3.14, post-pandemic mean 3.86).

In the second part of the survey (Figure 8), the users generally expressed a strong preference for the Library, with over 80% (36/44) users being "somewhat likely" or more to come back, and over 90% of users (41/44) being "somewhat likely" or more to recommend this to a friend. Users generally felt that it did replicate the feel of real libraries and bookshops to some extent, with over half (25/44) noting that it felt similar or very similar. Similarly, around half of participants found it "easier" or "much easier" to find books compared to bookshops, libraries (23/44) and online destinations (21/44), and felt that it contained "more" or "significantly more" new and interesting books compared to bookshops, libraries (23/44) and online destinations (24/44). Overall, survey respondents were very positive about the library and its major features.

### 5 DISCUSSION

Through our online deployment and survey, we were able to gather valuable feedback from readers and book browsers. In the open

feedback column, we had several useful insights generated by users.

**User Interface**: Three users wished for a mobile version in the feedback, with one noting "Mobile version must be created and [published] on App stores immediately", indicating the modern importance of smartphone-friendly interfaces. Indeed, although our interface does work on some mobile browsers, it does not on many older browsers due to a lack of WebGL support. We expect this limitation will improve in the future as newer devices generally do support WebGL. Two users commented that they would have liked to change the colour scheme: one user remarked that it was too dark, while another user wanted a "night mode" to make it even darker.

**Search Performance**: Our project primarily focused on book-browsing, rather than search, and consequently our search feature is much simpler than e.g. Google or Amazon's search functionality. Users commented that this could be improved. Four users noted that search could be improved: providing better search capabilities (such as advanced search by author/title/subject fields), improving the presentation of results (e.g. moving favorite or recommended books to the top), improving the discoverability of category or tag search, and ordering works by popularity or relevance. Recommendations, in particular, are interesting as they must function differently in a public library setting (with limited or no access to prior user data) compared with companies like Amazon which have vast access to prior user preferences via tracking and search history.

**Browsing Capabilities** Users generally praised the browsing and search features of the Library. One user noted that they were able to find "5 books on music theory in 5 minutes", while another noted that "It has everything I am looking for". Users also had suggestions for improving the experience further: one commented that they would have liked even more physicality in the form of different rooms/areas to browse around in, while another suggested a downloadable version for organizing their own books.

### 6 CONCLUSION

We have presented the design of the Library of Apollo, a virtual-reality library implemented in a web-browser application and designed to support library-like browsing and discovery. Our system was deployed publicly and attracted over two hundred visitors and 44 survey responses; our survey results suggested high user affinity for the book-browsing experience and library capabilities. Through the combination of features that we designed, we have built a virtual library which lends itself to serendipitous browsing and discoveries across a large, connected book dataset.

## ACKNOWLEDGMENTS

Anonymous for review.

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
