# OpenReview forum: "Library of Apollo: A Virtual Library Experience in your Web Browser"
_graphicsinterface.org/Graphics_Interface/2021/Conference/Second_Cycle — Reject_

### Official Review · Reviewer_MyLv · 2021-04-25
**Review of "Library of Apollo: A Virtual Library Experience in your Web Browser"**

**Rating:** 8
**Confidence:** 3

**Review:**

The paper presents the Library of Apollo, a virtual library implemented in a web browser application. It aims to bring together the connectedness and navigability of digital collections and the familiar browsing experience of physical libraries. Library of Apollo is an infinitely scrollable array of shelves that has 9.4 million books distributed over 200,000 hierarchical LCC categories. The system was deployed publicly and attracted over two hundred visitors and 44 survey responses. Survey results suggested high user affinity for the book-browsing experience and library capabilities.


Overall, I enjoy reading this paper. It targets a virtual library tool that can offer the same browsing experience and serendipitous discovery that physical libraries do. The paper is well-executed, well-written, and easy to follow. The presentation of the tool is clear and thorough. Here I only list a few minor concerns that I hope authors can address in their revision if the paper gets accepted.


**System Design Space and Rationales**

Although the authors did a good job in presenting the system, it is still helpful to understand and discuss other potential design options and provide rationales to support the current design. What is the design space of an online virtual library? What else we can do to mimic the “physical” library experience? Which part of the experience that users like the most when they are reading/browsing in a real library? Rather than simply presenting the system, I would suggest providing more deep discussions of some selected system features.


**Implementation Details**

Authors might need to provide more implementation details of the current system, such as how the data are stored, how to handle lots of user requests… Figure 6 is a good overview of the system’s structure, but it might be necessary to include more descriptive system details. It would be great if this project can be open-sourced. I believe the community can really benefit from this project.


**Improve the System**

Based on the users’ feedback, to me, it is also important to add more discussions about how to improve this system in the future. What potential features can be added in the next iteration of the system? How can we solve the “issues” brought up by the users?


**Summary**

Although I pointed out a few concerns above, I am still positive about this work. I think the online library tool will inspire future work in the field. Researchers and developers will benefit if this tool can be open-sourced.

---

### Official Review · Reviewer_7eXy · 2021-05-03
**Interesting system targeting a well-articulated problem, but limitations in the literature review weakens the scholarly contribution.**

**Rating:** 6
**Confidence:** 3

**Review:**

This paper presents a 3D virtual library aimed at recreating the capabilities physical libraries offer in supporting browsing and serendipitous discovery. The need to support browsing and serendipity have been well argued in the library and information studies, and the paper does a good job of summarizing and articulating this as well as the challenge of providing this support in digital collections. The paper then goes on to present a system and makes a decent case for how the proposed features address the concerns and stand to provide better support browsing and serendipity. The evaluation is light – a survey confirm that users of the system rated it positively – but sufficient for supporting what is primarily a systems contribution and presented honestly.

My one major concern is that the paper doesn’t sufficiently cover the ways in which browsing and serendipity have been covered by the field of information studies (IS). The literature review draws heavily from the IS literature to establish the problem space but in describing potential solutions, notes only two papers, both drawn from the HCI community. As an HCI researcher myself, I’m not overly familiar with digital librarianship literature, but it seems unlikely that 20+ years after identifying this as a problem there have been zero attempts within that community towards a solution. This paper may very well advance the state-of-the-art; however, without a meaningful review of the existing literature it will be hard for the paper to make a concrete contribution to the scholarly community.

My brief attempt at reviewing this literature space notes the following related works:

1.	B. Rogers, S. J. Cunningham and G. Holmes, "Navigating the virtual library: a 3D browsing interface for information retrieval," Proceedings of ANZIIS '94 - Australian New Zealnd Intelligent Information Systems Conference, 1994, pp. 467-471, doi: 10.1109/ANZIIS.1994.397010.
2.	Lemos J., Finn E. (2019) Babel VR: Multimodal Virtual Reality Environment for Shelf Browsing and Book Discovery. In: Stephanidis C., Antona M. (eds) HCI International 2019 – Late Breaking Posters. HCII 2019. Communications in Computer and Information Science, vol 1088. Springer, Cham. DOI: 10.1007/978-3-030-30712-7_5
3.	Almeida R., Cubaud P., Dupire J., Natkin S., Topol A. (2006) Experiments Towards 3D Immersive Interaction for Digital Libraries. In: Pan Z., Aylett R., Diener H., Jin X., Göbel S., Li L. (eds) Technologies for E-Learning and Digital Entertainment. Edutainment 2006. Lecture Notes in Computer Science, vol 3942. Springer, Berlin, Heidelberg. DOI: 10.1007/11736639_168
4.	Ilhan Aslan, Martin Murer, Florian Primessnig, Christiane Moser, and Manfred Tscheligi. 2013. The digital bookshelf: decorating with collections of digital books. In Proceedings of the 2013 ACM conference on Pervasive and ubiquitous computing adjunct publication (UbiComp '13 Adjunct). Association for Computing Machinery, New York, NY, USA, 777–784. DOI: 10.1145/2494091.2497318
5.	Pierre Cubaud, Pascal Stokowski, and Alexandre Topol. 2002. Binding browsing and reading activities in a 3D digital library. In Proceedings of the 2nd ACM/IEEE-CS joint conference on Digital libraries (JCDL '02). Association for Computing Machinery, New York, NY, USA, 281–282. DOI: 10.1145/544220.544282
6.	Pierre Cubaud, Pascal Stokowski, Alexandre Topol. 3D Metaphors to access a Digitalized Library. Virtual Reality Int. Conf.rence 2002, Laval, France, Jan 2002, X, France. ⟨hal-01124703⟩
7.	Matthew Jervis and Masood Masoodian. 2013. Visualization of physical library shelves to facilitate collection management and retrieval. In Proceedings of the 5th ACM SIGCHI symposium on Engineering interactive computing systems (EICS '13). Association for Computing Machinery, New York, NY, USA, 133–138. DOI:10.1145/2494603.2480321
8.	Julien, C., Guastavino, C., & Bouthillier, F. (2012). Capitalizing on Information Organization and Information Visualization for a New-Generation Catalogue. Library Trends 61(1), 149-161. doi:10.1353/lib.2012.0022.
9.	Beheshti, J., Large, A., Julien, C.‐A. and Tam, M. (2010), A comparison of a conventional taxonomy with a 3D visualization for use by children. Proc. Am. Soc. Info. Sci. Tech., 47: 1-9. DOI: 10.1002/meet.14504701061
10.	Karen Detken, Carlos Martinez, and Andreas Schrader. 2009. The search wall: tangible information searching for children in public libraries. In Proceedings of the 3rd International Conference on Tangible and Embedded Interaction (TEI '09). Association for Computing Machinery, New York, NY, USA, 289–296. DOI: 10.1145/1517664.1517724
11.	Cook, M. (2018). Virtual serendipity: Preserving embodied browsing activity in the 21st century research library. Journal of Academic Librarianship, 44(1), 145–149. DOI: 10.1016/j.acalib.2017.09.003


In addition, the following offers a set of requirements for online browsing support that could be used as a framework to articulate the contribution offered by Apollo:
* McKay, D., Buchanan, G. and Chang, S. (2018). It Ain't What You Do, It's the Way That You Do It: Design Guidelines to Better Support Online Browsing. In L. Freund (Ed.), Proceedings of the Association for Information Science and Technology (pp. 347– 356.) Hoboken, NJ: Wiley. DOI: 10.1002/pra2.2018.14505501038

Not all of the above-mentioned articles tackle exactly the same problem or explore the exact type of solution (though [8] does seem to be quite similar aside from the 10-year gap in 3D visualization technology available). But I’m not trying to suggest that the work here doesn’t make a novel contribution. The point is that there is a lot more work on this topic than the paper suggests and that this work should be synthesized to give the reader a better understanding of where the current work sits relative to existing body of literature. Only with that can the reader understand the work presented here and appreciate what it offers to advance scholarly knowledge on the topic.

Overall, I lean slightly towards acceptance, given the instructions to reviewers that emphasize finding reasons to accept papers. While I argue that the lack of connection to prior work risks limiting the potential contribution this paper might make, I don’t think this represents a fatal flaw. Put simply, I don’t believe this paper risks harming scientific progress on this topic, only that it risks being inconsequential to advancement in this space (or at least much less consequential that it could be if it more clearly articulated its contribution relative to the field as a whole). In as much as possible within the review cycle, I encourage the authors to expand out the literature review and better connect the design decisions and contributions of this paper to prior work.

---

### Official Review · Reviewer_rxk9 · 2021-05-04
**Reasonable web implementation but contribution is unclear**

**Rating:** 4
**Confidence:** 4

**Review:**

The paper's claimed contributions are "the design and implementation" of the LoA system, and the results of a survey from a public deployment. However, there is little in the paper to back up either of the claims in light of other systems that do similar things (both in virtual-library systems and in browsing systems more generally).

First, the paper states that it has made design improvements in terms of "increasing cross-connectivity", "ease of use", "navigability", and technical scale. However, there is no clear indication that previous work has failed in these regards - and there is substantial previous work that has looked specifically at serendipity and browsing. The submission does not clearly set out what has been done previously in terms of design dimensions for browsing an information space, and so it is not clear what is new or different in terms of the submission's design-space exploration that has not been considered in the many previous papers in Information Retrieval and Library Science. For example:
- Cook, M. (2018). Virtual serendipity: preserving embodied browsing activity in the 21st century research library. The Journal of Academic Librarianship, 44(1), 145-149.
DOI:https://doi.org/10.1145/3282894.3282927
- Beale, R. (2007). Supporting serendipity: Using ambient intelligence to augment user exploration for data mining and web browsing. International Journal of Human-Computer Studies, 65(5), 421-433.
The 3D interface looks good, but again there is not a clear context of what has been studied previously and how the current research goes beyond prior work. There have been many systems that provide a 3D interface to library content:
- Cook reference above
- Matti Pouke, Johanna Ylipulli, Ilya Minyaev, Minna Pakanen, Paula Alavesa, Toni Alatalo, and Timo Ojala. 2018. Virtual Library: Blending Mirror and Fantasy Layers into a VR Interface for a Public Library. In Proceedings of the 17th International Conference on Mobile and Ubiquitous Multimedia (MUM 2018). Association for Computing Machinery, New York, NY, USA, 227–231.

Second, the implementation of the IoA system seems to work well, and it is nice to see that it works in a standard web browser. However, there are again several issues and features that have been explored in previous work that are not replicated or discussed here, for example with respect to visual differences in the visual representations of the books. The representations in IoA seem to be fairly generic compared to other systems (e.g., Cubaud et al's VRML work in 1998, or Kleiner et al. Blended Shelf, both of which used realistic textures to differentiate the books' visual representations).
- Cubaud, P., Thiria, C., & Topol, A. (1998, May). Experimenting a 3D Interface for the Access to a Digital Library. In Proceedings of the third ACM conference on Digital libraries (pp. 281-382).

Third, the evaluation of the system does not provide evidence that the IoA system makes a real contribution. Although the comments gathered about the system are positive, there was no comparison or baseline condition, and certainly no comparison against the features and designs that have already appeared in literature. Therefore, it is impossible to determine if any of the design decisions made (e.g., to improve cross-links) were effective or successful in achieving their goals.

Overall, the paper does not provide a clear separation from previous work, and does not provide evidence that any real progress has been made in terms of research contribution.

---

### Meta-Review · Area_Chair_h4Pw · 2021-05-06

**Recommendation:** Reject
**Confidence:** 4

**Metareview:**

This paper has reviews that are spread across the scoring spectrum - one reviewer is quite positive about the work, one is in the middle, and one is much more negative. However, two of the reviewers largely agree on the weaknesses of the paper - the difference is more in terms of the reviewer's overall score rather than their assessment of the research. Both of the more negative reviewers highlighted the lack of a strong research contribution in light of the substantial previous work that has been carried out; and while one reviewer still sees value in the resulting system, the other questions whether there can be a research contribution without setting the work into the context of what has already been done.
This is a substantial limitation of the paper - a description of a system is not enough to provide a contribution to HCI research, and the reviewers do not see anything that stands apart from what we already know. The main features of the system have been considered in previous work, and no attempt is made to compare to these earlier systems or explain how the new system goes further.
Overall, this limitation is too large to allow us to argue for acceptance.

---

### Decision · Program_Chairs · 2021-05-08

Reject